# HTTP Extensions for the Management of Highly Dynamic Data Resources

✉ Lars Gleim[1] ⓘ, ✉ Liam Tirpitz[1] ⓘ, and Stefan Decker[1,2] ⓘ

[1] Databases and Information Systems, RWTH Aachen University, Germany
[2] Fraunhofer FIT, Sankt Augustin, Germany

**Abstract.** As Semantic Web Technologies are increasingly employed for the management of highly dynamic data resources, e.g., the Industrial Internet of Things, resource versioning, state synchronization and distributed data management infrastructures are gaining practical relevance. The HTTP Memento protocol has recently been discussed as a promising building block for the implementation of such services for Findable, Accessible, Interoperable and Reusable (FAIR) Data. While this standard already enables the management and discovery of persistent, immutable and versioned resources on the Web and in Knowledge Graphs, it lacks support for the management of data updated at high frequencies and only provides inefficient means for managing resources with many revisions.

To address these shortcomings, we propose three extensions to the HTTP Memento protocol: arbitrary resolution timestamps, resource creation support and range requests for TimeMaps. We provide a reference implementation of our proposals as open source software and quantitatively evaluate the extensions' performance, showcasing superior results in terms of resource capacity, insertion correctness, latency and amount of transferred data. Based on a qualitative analysis, we conclude that in conjunction with our proposed extensions, the HTTP Memento protocol addresses a variety of data management challenges including data *archiving*, *citation*, *retrieval*, *discovery*, *synchronization* and *sustainability* for highly dynamic data on the Web and in Knowledge Graphs, providing a promising foundation for prospective standardized and interoperable data management solutions.

**Keywords:** HTTP Memento protocol · FAIR Data Management · Decentralization · Version Management · State Synchronization. · Linked Data · RDF.

## 1 Introduction

Resources on the Web evolve over time and some resources change faster than others. Handling such resources can be problematic because data cannot be referenced and cited reliably if it changes or disappears altogether [8]. To combine the opposing requirements for dynamic resources and reliable citations, a suitable versioning and persistent identification mechanism is needed, that allows to reliably capture, identify and retrieve individual, immutable resource revisions. While semantic data management (SDM) and versioning solutions in the Semantic Web community are largely using SPARQL, LDP [22] or plain HTTP as their primary access mechanism, there is no standard for managing highly dynamic resources that would allow the handling of RDF data analogously to any other resource on the Web, as promoted e.g., by the FAIR principles for

scientific data management [27]. Especially in the IoT context a joint and standardized mechanism to handle highly dynamic data resources is missing.

The HTTP Memento protocol [20] has been successfully employed for time-based resource access and identification in the context of SDM [9,15,21] and could provide the basis for interoperable solutions, tightly integrated with the HTTP protocol and therefore the core technologies of the Web. However, in its current form, the Memento protocol was created for applications with slowly changing data in mind, such as traditional websites or library resources [20]. As such, Memento does not perform well in scenarios with highly dynamic data resources, because it is limited by design decisions like the use of RFC1123 timestamps [4] with a maximum resolution of one second, limiting the frequency of data changes that can be handled. Nevertheless, the need for standardized identification and retrieval of resource revisions also exists in applications with highly dynamic data resources, for example in the Industrial Internet of Things, where a large variety of different data elements, like sensor and machine data, must be captured at high frequencies [17]. A prominent example is provided by the recently standardized W3C Web of Things API, which promotes, e.g., the direct exposure of current sensor readings through web resources [14]. In such a scenario, a single data resource may describe the state of a machine or sensor, which changes multiple times per second. While stream processing [6], as well as data propagation and notification systems [5], have been actively investigated in recent years, the unified management and identification of individual data points received little attention. Instead, efforts such as the JSON Time Series data format [2] focus primarily on providing a lightweight data-interchange format. At the same time, different approaches for versioning semantic data were developed [16], but those often focus explicitly on RDF data and cannot provide straightforward interoperability with existing Web resources. However, especially in the context of industrial use case scenarios, each individual state of such a resource may need to be persistently identified and retrievable [10,11]. The Memento protocol is a promising candidate to provide these services, however, currently inadequate to handle such highly dynamic resources. Therefore, we propose and evaluate extensions to the existing Memento protocol with the goal to provide a standardized mechanism to create, access and identify revisions for highly dynamic data resources.

**Contributions.** Based on our discussion of shortcomings of the Memento protocol, we propose three extensions to the HTTP Memento protocol:

- An updated datetime format, allowing arbitrary resolution timestamps, to uniquely identify individual resource revisions, even at high frequencies.
- Support for Memento creation as part of the protocol, to allow clients to reliable cite the resource revisions they created.
- Temporal range requests for TimeMaps, enabling the targeted retrieval of specific temporal ranges of Mementos for a more efficient discovery, especially for resources with large numbers of revisions.

We analyze the practical benefits of our proposals in a both quantitative and qualitative evaluation, concluding them to provide a promising foundation for the standardized management of highly dynamic data resources on the Web.

**Paper Organization.** The remainder of this paper is structured as follows. Section 2 provides an overview of corresponding related work and fundamental technologies.

Section 3 discusses the proposed extensions and their benefits in detail. Section 4 evaluates the proposed extensions using our open source reference implementation. Finally, we conclude our work in Section 5.

## 2 Related Work

In the following, we provide a short overview of prior work towards data versioning and temporal data management on the Web. We then introduce the Memento protocol and discuss its applications in data management on the Web.

**WebDAV.** As summarized by Whitehead [26], the WebDAV protocol and its extension DeltaV provide capabilities for remote collaborative authoring, metadata management, version control, and configuration management of Web resources. Extending upon HTTP, WebDAV adds operations for overwriting prevention, properties, and namespace management, while DeltaV builds upon WebDAV to offer versioning (checkout and checkin), autoversioning, workspaces, activities, and configuration management. Although both WebDAV and DeltaV are IETF Web standards, their practical adoption remains low to date, in part due to the general complexity of the WebDAV protocol. While Tim Berners-Lee still proposed the usage of the protocol for resource management in his 2009 vision of *Socially Aware Cloud Storage* [3], the later implementation *Solid* [18] instead implements the Linked Data Platform specification [22] standardized by the W3C in 2015.

**The Memento Protocol.** A more recent and much simpler approach to resource versioning is provided through the HTTP Memento protocol [19], which enables the retrieval of *Mementos* – historic states of resources – via time-based HTTP content-negotiation. As summarized by Gleim and Decker [9], the Memento framework distinguishes four logical components: Original Resource, Memento, TimeGate, and TimeMap. A Memento $\langle u, t \rangle$ captures the state of an Original Resource with URI $u$ at a given point in time $t$ (exposed via the `Memento-Datetime` HTTP header). Mementos are intended to be *immutable* and may optionally be associated with one or more distinct Memento URI(s) for *referenceability*. Such a URI-M must further identify the URL of its Original Resource in an HTTP `Link` header. Using the `Accept-Datetime` HTTP request header, historic states of Original Resources may then be requested from a so-called TimeGate through time-based content-negotiation, and are serviced through either a direct HTTP response or HTTP redirection to external archive locations, providing a simple solution to the archiving problem.

Additionally, the Memento protocol also enables revision discovery and synchronization through TimeMaps, which provide a listing of available Mementos $\langle b, t* \rangle$ at points in time $t*$ for a given Original Resource $b$, i.e., a history of available revisions with respective associated distinct Memento URIs. Thus, exposing up-to-date TimeMaps for resources enables trivial change monitoring and the discovery, retrieval, and thus synchronization of resource state. Depending on the application scenario, TimeGate, TimeMap, and/or Mementos may all be provided by the Original Resource provider. It is, however, similarly possible to deploy all components independently of each other, as well as with optional redundancies. Notably, third parties may also provide archiving services by storing Mementos and/or providing lookup services (i.e., an *external*

TimeGate) for resources from other domains, such as already provided by archive.org. Thus, adoption does not hinge on the support of any individual group or organization but may be adopted by interested users in backward compatibility with existing resources on the Web. Individual Mementos may further be resolved to multiple URI-Ms, i.e., different storage locations, (e.g. via TimeMaps), supporting explicit redundancy. Gleim and Decker [8] conclude that the Memento protocol provides a promising solution for the *archiving*, *citation*, *retrieval*, *discovery*, *synchronization* and *sustainability* challenges of data management. Further details, including discovery procedures for TimeGates, Mementos and Original Resources, can be found in RFC7089 [19].

**Data Management with the Memento Protocol.** In the following, we shortly introduce prior work exploring the application of the Memento protocol in the context of data management solutions. Meinhardt et al. [15] proposed a system for the management of Linked Data enabling access to arbitrary historical data states through the Memento protocol. The authors further implemented a custom REST API to enable incremental updates to the underlying RDF data. Due to its reliance on custom API endpoints, the approach is however not well suited for applications with arbitrary Web services. Verborgh [25] described an approach employing the Memento protocol to query historical datasets using the SPARQL query language. Taelman et al [23] then adopted the approach for historical access to linked data through the Triple Pattern Fragments API. Extending upon these approaches, Vander Sande [24] proposes an integrated data publishing solution for libraries, providing historical access to Linked Data through various access mechanisms. Anderson [1] proposed the Dydra graph store which implements temporal RDF dataset versioning analogously to the Memento TimeGate pattern. Recently, Gleim et al. [11] employ the Memento protocol for resource versioning in the implementation of the semantic data management system FactStack [11] based on the FactDAG data interoperability model [10]. Nevertheless, the Memento protocol's shortcomings w.r.t. highly dynamic data resources remain unaddressed to date.

## 3   Optimizing the Memento Protocol

In the following, we propose multiple, independent extensions to the existing Memento protocol, with the goal to enable support for highly dynamic data resources. First, we introduce an exemplary industrial use case scenario to highlight the need for Memento protocol extensions to better support highly dynamic data resources. Motivated by this use case, we propose the adoption of the RFC3339 datetime format [13] which supports arbitrary time resolution, instead of the currently used RFC1123, which is limited to a resolution of one second. Subsequently, we propose to broaden the scope of the Memento protocol beyond pure data retrieval and extend it to support the creation of resources and their revisions via `PUT` and `POST`, as well. This especially enables clients to reliably and persistently cite individual resource revisions immediately upon creation. Finally, we propose a ranged request for TimeMaps, which enables clients to retrieve arbitrary sections of a TimeMap with a single request, reducing the communication overhead associated with large TimeMaps commonly associated with highly dynamic resources.

**Use Case.** To highlight the need to extend the Memento protocol towards highly dynamic resource support, we introduce the example use case illustrated in Figure 1.

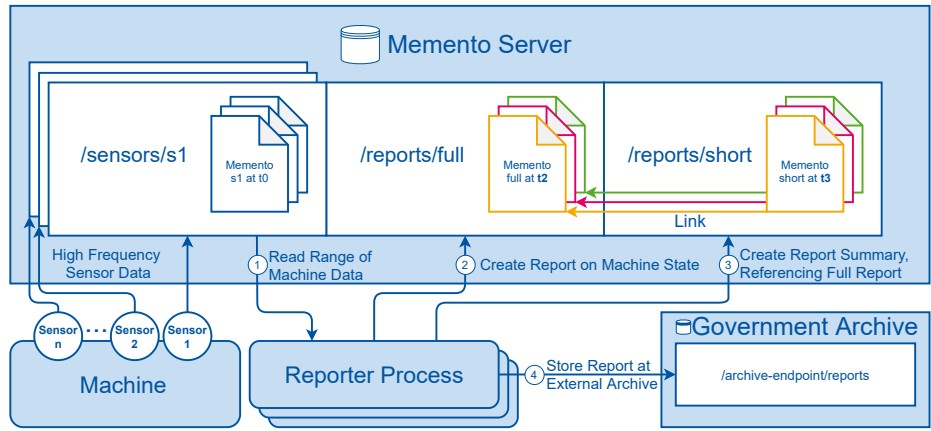

**Fig. 1.** An example highlighting the need for HTTP extensions to handle highly dynamic resources.

We will refer to this use case in the following sections to motivate individual Memento extensions. Our use case considers a production machine in an industrial IoT setting, which can be observed via multiple sensors writing their data to a Web server via HTTP `PUT` or `POST` requests with high frequency, possibly following the Linked Data Platform specification [22]. In the following, we refer to this Web server as the *Memento Server*. Each sensor writes to a single resource and each revision of such a resource (stored and subsequently retrievable as a Memento) reflects a specific sensor value, i.e, the state of a sensor at a specific point in time. The use case also consists of multiple reporter processes. These software components consider a certain range of Mementos associated with certain sensor resources to create reports indicating the state of the machine. A reporter process first creates a full report with all relevant information, by updating a resource on the Memento server. The reporter process also creates a short summary of this report directly after the full report is created. To maintain useful provenance information, each summary also references the full report it was generated from. Finally, since the machine produces safety-critical parts, regulated by the government, the summary of the report is also stored externally on a government archive server.

### 3.1 Changing the Datetime Format

The Memento protocol relies on timestamps to provide access to naturally ordered resource revisions via datetime negotiation or TimeMaps. These timestamps follow the RFC1123 [4] format, which provides a static resolution of one second. However, in industrial applications like the use case illustrated in Figure 1, we consider sensors updating resources multiple times per second, each change leading to a Memento that needs to be identifiable. The RFC1123 timestamps cannot uniquely identify multiple revisions that occur within the same second and timestamp collisions are unavoidable for high-frequency data. Therefore, we propose a revision to the Memento protocol that changes the format of the timestamps to the more modern and flexible RFC3339 [13]. RFC3339 standardizes the use of datetime formats based on ISO 8601 [12] for use in

internet protocols and allows the representation of fractions of seconds with arbitrary precision. The use of RFC3339 in the Memento protocol would allow the natural ordering of resource revisions with an arbitrarily small distance to each other. While computer systems only provide timestamps with a finite resolution, the RFC3339 format would allow the protocol to be used on current systems with the highest possible resolution, as well as with future systems, which may provide an even higher timestamp resolution. Additionally, for most applications, the uniqueness of a timestamp is more important than its actual accuracy. Therefore, even systems with lower resolution timestamps can profit from the additional resolution allowed by the RFC3339 format by simply using the least significant digits as a counter to guarantee the uniqueness of identifiers assigned in the same timestamp interval. While these virtual timestamps are not accurate timestamps up to the least significant digits, they provide uniqueness and natural ordering while still expressing time information with the highest possible precision. Since RFC3339 allows arbitrary fractions, the size of the counter can be chosen to fit the needs of the application. For the considered use case, the use of RFC3339 would allow the sensors to send with an arbitrary frequency, without losing the unique identifiability of individual Mementos.

### 3.2 Considering Resource Creation

In our use case, we need to consider reporter processes, which revise two resources, the full report and its summary. The summary additionally links to the full report, establishing useful provenance information. To do so, the reporter process needs to learn the unique identifier of the full report, before it can create the summary. Additionally, there may be multiple reporter processes writing to the same resources. Using the Memento protocol, the unique identifier may be obtained by combining the URI of the resources with the `Memento-Datetime` [9], which is assigned by the server. Since the Memento protocol does not consider the creation of resources via HTTP `PUT` or `POST`, there is no standardized way for the client to learn the `Memento-Datetime` that was assigned to the Memento it created. While the most recent `Memento-Datetime` may be obtained using an additional `GET` request to the resource created using `PUT` or `POST`, this poses multiple problems. First, following each `PUT` or `POST` request with a `GET` request creates unwanted overhead. Additionally, since multiple clients may write to the same resource, race conditions may occur and it cannot be guaranteed that the returned `Memento-Datetime` is actually associated with the Memento created by the requesting client, as illustrated in Figure 2. In our use case, this would lead to a summary that references the wrong full report. Instead, we propose the extension of the Memento protocol to also consider the creation of Mementos through RESTful APIs by reusing the existing `Memento-Datetime` header as a response header in the context of `PUT` or `POST` requests. The server may then further indicate the location that the created resource may be retrieved from through the HTTP `Content-Location` header. Note, that this approach may be implemented in a backward-compatible manner with arbitrary REST endpoints (as long as they internally guarantee, that individual resource revisions are unique). The Memento protocol considers different scenarios where TimeGates, Original Resources and Mementos may exist distributed across different servers or may be handled by the same entity. We also consider different cases for the creation

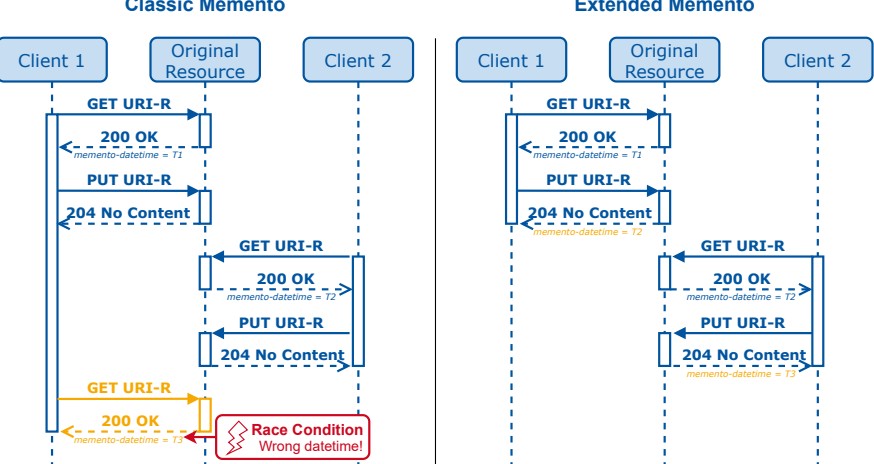

**Fig. 2.** The necessary requests for Memento creation with and without the proposed extension. The extension avoids additional requests and race conditions, which may lead to wrong datetime associations at the creating client, by returning the assigned `Memento-Datetime` directly with the response to a request.

of Mementos. However, we always assume that an Original Resource acts as its own TimeGate and assigns unique `Memento-Datetime`s to each resource revision. After a Memento has been created by the Original Resource it may be handed over to another Memento server for storage.

**Creating Mementos through the Original Resource.** In the centralized case, the Original Resource (OR) also acts as its own TimeGate. We propose the standardization of `PUT` and `POST` requests towards the Original Resource. To create a new revision of the Original Resource, a client may issue a `PUT` or `POST` request to the OR. If the request is successful, the current representation is updated to the new revision and a new Memento is created. The server communicates the unique identifier it assigned to the Memento via the `Memento-Datetime` header. While the classic Memento protocol already specifies the `Memento-Datetime` response header, its exact meaning is clear from the type of request. Used as a response to a `PUT` or `POST` request, the `Memento-Datetime` header communicates that the content of the request has been persisted with the returned, unique timestamp. This proposed additional response header allows clients to directly and reliably reference the resource revisions they created, as illustrated in Figure 2.

To achieve the desired immutability for reliable citations, a `DELETE` request may only create a tombstone object. The current representation of the resource would act like a deleted resource, while the existing Mementos remain available.

**Storing Mementos in an external Memento Archive.** Our example use case also requires the reporter process to hand over the summary reports to an external archiving service provided by the government. To store already existing Mementos at an external TimeGate, we propose the utilization of an archiving endpoint (URI-A). This archiving endpoint consists of the URI of the Memento server, but also encodes the URI-R of

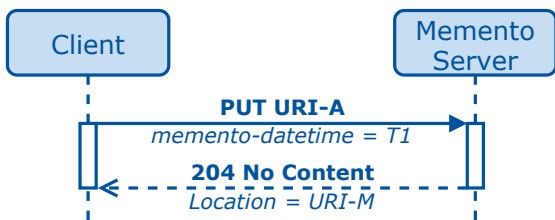

**Fig. 3.** The proposed process of storing an existing Memento at an archiving server that is not holding the associated OR. The client passes the `Memento-Datetime` as a header and encodes the location of the OR as part of the URI-A. The server responds with the new Memento location.

the Original Resource in its path. The archiving endpoint may exist at an arbitrary location on the server, which should be discoverable via a well-known location, e.g., `http://archive.tld/.well-known/memento-archive-location/`. At this location, clients can learn the actual archive endpoint, e.g. from a JSON representation such as {`"archive_location":"http://archive.tld/archive-endpoint"`}.

This way, Memento archiving services provide a clearly specified location that stores and provides access to Mementos belonging to other Original Resources, which may exist in parallel to Original Resources the archiving service maintains itself. Encoding the URI-R as part of the URI-A instead of a separate request header enables a unique endpoint for each OR which acts as a TimeGate for that OR and may even provide resource-specific human-readable information at that location, such as the TimeMap in HTML format. To store a Memento, we reuse the already existing `Memento-Datetime`, as illustrated in Figure 3. Used as a request header, the `Memento-Datetime` indicates the unique `Memento-Datetime` that the OR assigned to the Memento in the request at its creation. The archiving endpoint responds with the URI-M identifying the storage location of the Memento. Note, that `Memento-Datetime` is used as a request header instead of a response header. Similar to the creation of Mementos at the OR, the semantics of the header is implied by the type of the associated request.

The ability to store Mementos at external archiving services in a standardized way allows for the realization of simple push-based resource state synchronization mechanisms and redundant resource archiving, both of which are relevant challenges for data management solutions on the Web. With regard to our use case, the extension for resource creation allows the reporter processes to reliably reference the full reports from their summaries and to create synchronized copies of the summaries at an external archiving service.

### 3.3 Range Requests for TimeMaps

In the considered use case, sensors create resource revisions with high frequency. Therefore, a single resource is expected to have a large number of revisions. The reporter processes in this use case are only interested in a specific range of Mementos, i.e., the Mementos that were created since their last execution. The Memento protocol specifies TimeMaps, listing the Mementos for a resource known to the TimeGate that provides

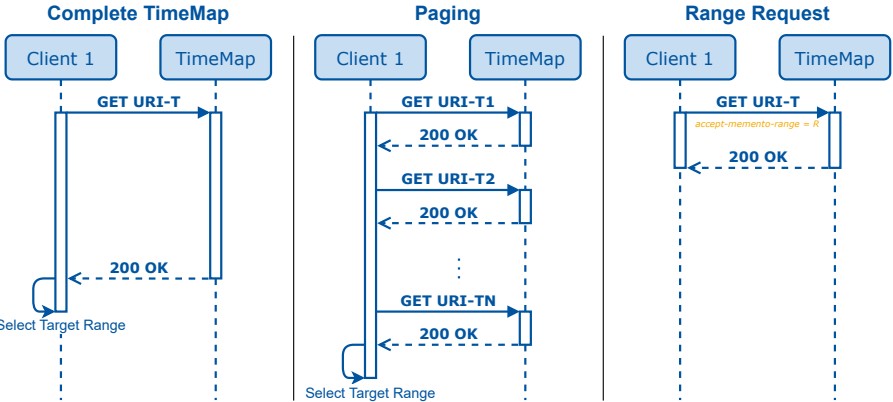

**Fig. 4.** Comparison of retrieval methods for TimeMap fragments. A complete TimeMap always returns every fragment with a single request, but for large TimeMaps, this is inefficient. Linearly paging through TimeMap fragments transmits fewer data per request, but may require many requests. The proposed range-request returns only the requested range of Mementos with a single request.

the TimeMap. If many Mementos exist, as is the case in our use case, this TimeMap may be long and difficult to handle. However, the clients in our use case are not interested in the full TimeMap, but only in a specific fragment.

The classic Memento protocol provides a paging mechanism, which divides the TimeMap into pieces that can be accessed separately, where each page also points to its predecessor and its successor. Since the URI format identifying the individual pages is not standardized, the TimeMap can only be paged linearly, without skipping ahead or even searching for a certain time range by performing a binary search. Therefore, if a client is looking for a range of Mementos at the end of the TimeMap, the complete TimeMap is downloaded and many requests are necessary to reach the desired part of the TimeMap, which creates a huge overhead for resources with many Mementos. Additionally, the client may need to combine the results from multiple pages locally and remove parts of the first or last page that do not match the target range. The Memento protocol also specifies Index TimeMaps, which only contain links to other TimeMaps and a datetime range that is covered by each linked TimeMap. Index TimeMaps can again point to other Index TimeMaps. However, Index TimeMaps are primarily intended for distributed TimeMaps across multiple archives. While this mechanism can be used to create tree structures of TimeMaps that guide clients to a specific range of Mementos without sending the complete TimeMap, it still requires the client to request multiple TimeMaps and assemble the results from these requests to obtain the intended range of Mementos. Therefore, Index TimeMaps are not considered further. Instead, we propose a new header that clients could use to request a specific range of a TimeMap, the `Accept-Memento-Range` header. Unfortunately, RFC3339 does not cover time periods, but they are specified by ISO 8601, for which RFC3339 is a profile. Therefore, we utilize the time period syntax of ISO 8601, more specifically its *explicit* syntax for periods that

connects two timestamps with a forward slash, e.g., *iso-date-time-start / iso-date-time-end*. Semantically, this indicates the time period between both timestamps. In a future standardization effort, the `Accept-Memento-Range` header may also be realized by creating a datetime unit for range requests following RFC 7233 [7].

In our use case, the reporter process may set the `Accept-Memento-Range` header with the desired time period of the presented format to request a fragment of a TimeMap which only covers the relevant time period. To maintain compatibility with legacy servers, a Memento client must be able to fall back to a regular TimeMap request, if the header is ignored by the server.

### 3.4   Ensuring Compatibility with Legacy Systems

While proposing changes to an existing and established protocol, the implications for the compatibility with the existing protocol revision also need to be considered. Most of the proposed changes can be implemented as an extension, without any impact on existing Memento implementations. The additional request headers for the creation of Mementos or the retrieval of TimeMaps would only be used by a client compatible with the Memento extension. A legacy server would simply ignore the unknown headers and a compatible client must handle a legacy response. Similarly, a modern server may choose not to implement individual extensions to keep its complexity low and a client also needs to handle any combination of enabled extensions. If the server uses extensions, incompatible clients would simply ignore the additional response headers that are returned for HTTP `PUT` or HTTP `POST` requests. Therefore, these extensions would not break the existing Memento infrastructure.

However, the datetime format used by the Memento protocol cannot be changed without breaking compatibility with legacy systems. To address this problem, the updated datetime format could be implemented as an extension as well, using an additional header, the `Memento-Version` header. If the client initiates a request with `Memento-Version=2`, the server has to use RFC3339 timestamps for its responses, if the extended Memento protocol is supported. This holds for TimeMaps as well as `Memento-Datetime` headers. If the client does not set this header, the server acts as a legacy server and uses RFC1123 for the datetime format in its responses. The use of RFC1123 timestamps may lead to ambiguous identification of Mementos, as previously discussed. In that case, the server may use an arbitrary but consistent reduction and ignore additional Mementos in interactions with legacy clients, e.g., the server may only list and return the first Memento that was created within a certain second. If this behavior is undesirable, the server may also reject legacy requests. If a client wants to use the RFC3339 datetime format for its requests, using the `Accept-Datetime` header, it may not know if the server supports that extension. Therefore, the protocol extension must ensure that such an interaction can fall back to classic Memento. This could be achieved by sending two request headers with the initial request. The `Accept-Datetime` header uses the RFC1123 format with reduced resolution and an additional header, `Accept-Datetime-2` for example, specifies the RFC3339 datetime. If the server does not support the extension, it will ignore the second header and the client needs to handle the legacy response, which can be identified by looking at the

format of the `Memento-Datetime`. Otherwise, the server supports the extension and the client may send subsequent requests without the legacy headers.

The proposed extensions further maintain HTTP idempotence of all requests. While range request and increased timestamp accuracy clearly have no impact on idempotence, creating Mementos through the Original Resource creates a new Memento with user-provided content and server-assigned `Memento-Datetime` for each `PUT`/`POST` request. It is therefore idempotent w.r.t the latest resource state itself, even though the header is updated. Storing Mementos in a third-party Memento Archive is always idempotent, both w.r.t. the resource state and its Memento header. Therefore, from the viewpoint of the "non-Memento-aware" Web, all described methods are idempotent.

Together, the proposed changes enable the Memento protocol to be used for data management in high-frequency environments, where every revision of a resource has to be captured, while also maintaining compatibility to legacy systems.

## 4   Evaluation

To evaluate the proposed extensions to the Memento protocol, we implemented a minimal Memento server as a Node.js (v14) application with an in-memory Redis backend (v6), which we released as open source software[3], and evaluate its performance using a single-node deployment of the server application and a client on a workstation with Intel i7-8700K CPU, 64 GB of RAM and NVMe SSD. The goal of the Memento server implementation used for the evaluation was not to provide a highly scalable application but to compare the proposed extensions with the classic Memento protocol. The repository also contains a written description of the implementation details.

We conducted two separate experiments. The first experiment evaluates the use of the RFC3339 datetime format, as well as the benefits of using Memento in the context of resource creation. The second experiment evaluates the proposed range request for the retrieval of Memento TimeMaps. All created resources in both experiments contain random strings of length 20. All plots reflect the averaged result over 10 repetitions of the associated experiment. The error bars indicate the 99% confidence interval.

### 4.1   Experiment 1 - Inserting Resource Revisions

First, the performance of the `Memento-Datetime` response header for `PUT` and `POST` requests to the OR is evaluated experimentally in combination with the updated timestamps in RFC3339 format. In this experiment, the OR acts as its own TimeGate. We assume an application in which the client needs to reference the exact resource revision it created and therefore needs to learn the unique `Memento-Datetime` assigned to its revision. Clients create revisions of a single OR using `PUT` requests to that resource, with different frequencies. The server creates a Memento for each request, assigns unique RFC3339 timestamps and directly returns them via the `Memento-Datetime` header with the response. An insertion is only considered successful if the server assigns a unique identifier for the inserted resource revision and the client learns the correct identifier so it can reference the associated resource revision.

---

[3] https://git.rwth-aachen.de/i5/factdag/memento-server

In the first variation of that experiment, the client ignores the returned `Memento-Datetime` header and issues a subsequent `GET` request to learn the `Memento-Datetime` header. This emulates a scenario with the use of high precision RFC3339 timestamps, but without the `Memento-Datetime` header for `PUT` or `POST` requests. In the second variation, the client considers the returned `Memento-Datetime` header to learn the assigned `Memento-Datetime` and does not issue a subsequent `GET` request. The correctness of insertions with the third variation, the classic Memento protocol with RFC1123 timestamps and without the `Memento-Datetime` response header is not determined experimentally. Instead, the best-case scenario of one correct insertion per second is plotted based on the availability of timestamps and the pigeon-hole principle. Note, that due to effects like jitter or processing delay, requests may collide for a unique timestamp in one second, while the timestamp provided by the following second is not used at all. Therefore, the classic Memento protocol may perform worse in practice, especially for comparably low frequencies of around 1 revision per second.

**Correctness of Insertions.** Figure 5a plots the percentage of successful insertions for different loads (insertions per second) on the server for the classic Memento protocol, the extended protocol and a hybrid with RFC3339 timestamps but without the resource creation extension and its response headers respectively. In the case of classic Memento, due to the limited timestamp resolution, only a single revision gets a unique identifier per second. That limits the effective load on a single resource to one insertion per second. Otherwise, most of the insertions cannot be uniquely identified. If RFC3339 timestamps are used, but the associated timestamp is not returned with the response to a `POST` or `PUT` request, the client must issue a subsequent `GET` request to learn the exact timestamp. However, between the `POST` request and the `GET` request, another insertion may have taken place on the same resource, as illustrated in Figure 2. In that case, the client learns a timestamp associated with the wrong resource revision. This is increasingly likely for high loads on a single resource. The client checks if the correct timestamp was returned by comparing the inserted resource content with the returned content. If the contents match, the insertion is considered successful otherwise the insertion failed. For the extended Memento protocol, the timestamp associated with an insertion can be learned directly from the response to a `POST` or `PUT` request and a subsequent `GET` request is not necessary.

**Performance.** The previously presented results show increased correctness for the creation of resource revisions with the extended Memento protocol, but the performance of such creation events is important as well. The experimental results in Figure 5b show, that the extended Memento protocol is faster for creation events if the client needs to learn the unique identifier of the created resource revision. While the performance of our Memento server implementation could likely be significantly optimized, performance degradation over a certain threshold is to be expected from any implementation. The significant performance drop observed above roughly 1500 requests per second does not hinder the direct comparison of the protocol performance itself. The plot shows that the used implementation can handle twice as many creation events per second if the extended Memento protocol is used, before the time to completion increases notably. These advantages are due to the need for a client to send a subsequent `GET` request to learn the most recent `Memento-Datetime`. Therefore, each creation event consists of

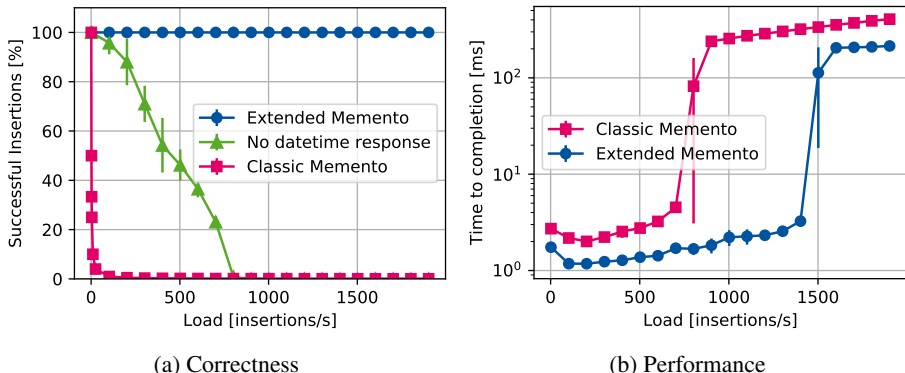

(a) Correctness                                  (b) Performance

**Fig. 5.** Results of Experiment 1, comparing the extensions to the classic Memento protocol regarding correctness and performance of citable resource revision insertions.

a `PUT` or `POST` request, followed by a `GET` request. This roughly doubles the load on the server and increases the time to complete the operation for the client, because two sequential requests need to be issued to the server. Since we execute requests against localhost, the impact of network latency on our results is negligible. Increased network latency would impact the traditional Memento protocol (with one or multiple requests) equally or more than our proposal (with only a single request for both Memento creation and TimeMap retrieval). An evaluation with larger resources (compared to the current 20 character strings) would effectively add the same transmission time offset for either protocol version since this affects the initial PUT/POST request time equally for both protocol versions. Subsequently, the relative overhead of the second request diminishes as larger resources are transferred. Nevertheless, the absolute overhead remains effectively unchanged, since an additional round trip is needed in the traditional protocol.

The evaluation of this experiment shows that the use of RFC3339 timestamps with Memento allows the assignment of unique identifiers to resource revisions even for highly dynamic resources, which is not possible with the currently used RFC1123 timestamps. The experiment also shows that timestamps with a high resolution alone are not sufficient in applications that require the creating client to reference the exact revision it created. Instead, the protocol also needs to consider the creation of resource revisions and directly notify the client of the unique identifier that was assigned to the revision. With both proposed protocol extensions combined, the correct assignment and referencing of resource revisions is guaranteed even for highly dynamic data resources, while reducing the overhead generated by additional requests in applications that require reliable references to specific resource revisions.

## 4.2 Experiment 2 - Accessing TimeMaps

The second experiment considers the proposed extension to the retrieval of TimeMaps, namely the `Accept-Memento-Range` header. To evaluate its performance, and compare

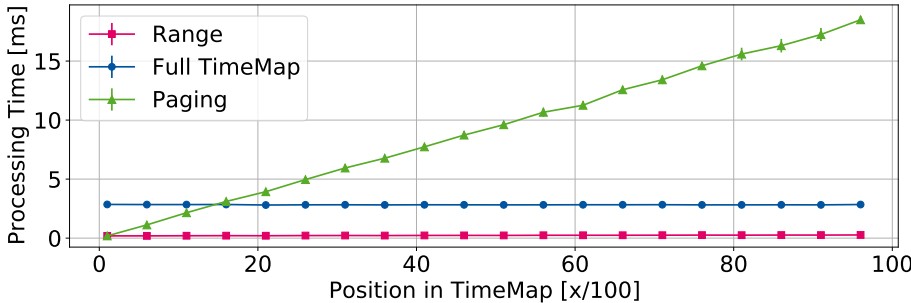

**Fig. 6.** Processing time for retrieving 100 consecutive entries in a varying position from a TimeMap with 10 000 entries. Retrieving the whole TimeMap always requires the same time, but produces overhead, because the TimeMap is larger than the target range. Linearly paging through the TimeMap with 100 entries per page varies in its duration based on the position of the target range. The proposed datetime-range request results in a consistently low processing time for the client.

it to the already standardized methods of retrieval, we create a resource with 10 000 revisions and set up a client to retrieve a subset of revisions based on a given datetime-range which includes 100 Mementos. The position of the targeted Mementos is varied with each execution so that the first execution needs to isolate the entries on positions 0 to 99 and the last execution targets the entries on positions 9900 to 9999 of the TimeMap. The experiment is executed for the retrieval of a full TimeMap, for the retrieval via a paged TimeMap and for retrieval via the proposed range request, respectively. For each execution, the processing time is measured. In the case of a full TimeMap, that includes the time for the request and the response itself, as well as the time the client spends to identify the targeted elements from the TimeMap. Similarly, in the case of a paged TimeMap, the processing time includes the time for the individual requests, as well as the time the client needs to evaluate if a page lists some or all of the targeted Mementos and the time to create the final list, which may consist of fragments combined from multiple pages. The results plotted in Figure 6 show that the processing time to retrieve a segment of a TimeMap is constant if the full TimeMap is retrieved and the desired segment is isolated by the client. On the other hand, the processing time for the retrieval of a segment via a paged TimeMap depends on the position of the segment within the TimeMap. If the segment is at the beginning of the TimeMap, the paged approach may be faster than the retrieval of the complete TimeMap. However, since the client needs to page through the TimeMap page by page, the processing time increases linearly if the position of the target range is moved towards the end of the TimeMap. If the target range is at the end of the TimeMap, the processing time for a paged TimeMap is considerably higher than for the retrieval of a full TimeMap, because increasingly many pages need to be requested. Like the retrieval via a complete TimeMap, the processing time for retrieval via a range request is independent of the position of the target range within the Map and can be completed with a single request. Since the transmitted amount of data may be considerably smaller depending on the relation between map size and range size (1/100 of the user data with the chosen example) the processing time for a range request

is lower compared to the retrieval of the full TimeMap. While the retrieval via pages may have similar processing times, this is only possible if the desired range is towards the front of the Map.

Note, that the exact results heavily depend on the size of the TimeMap and the size of the target range in relation to the size of the TimeMap and this experiment only considers a single combination of those parameters. However, the provided data does clearly show how the range request for TimeMaps can have a positive impact on the communication overhead and the processing time for the client. At the same time, depending on the implementation, the range request may increase the computational load on the server, especially compared to statically cached TimeMap pages. Since the extension is optional for the server, it may decide to deactivate this extension if computational resources are limited.

## 5  Conclusion

In this work, we proposed three independent extensions to the HTTP Memento protocol to address its current shortcomings with respect to the management of highly dynamic data resources, such as increasingly prevalent in the Web through the influence of Industrial Internet of Things technologies. We specifically propose the following modifications: a) An updated datetime format, allowing arbitrary resolution timestamps, to uniquely identify individual resource revisions, even for highly dynamic resources. b) Support for Memento creation as part of the protocol, to allow clients to reliable cite the resource revisions they created. c) Temporal range requests for TimeMaps, enabling the targeted retrieval of specific temporal ranges of Memento TimeMaps for the more efficient discovery of resource revisions, especially for highly dynamic resources with large numbers of resource revisions.

Based on respective quantitative performance evaluations and qualitative analysis in the context of a concrete usage scenario in the context of industrial sensor data management, we demonstrated the superior performance of all three proposals for the management of highly dynamic data resources compared to the plain Memento protocol. Notably, we were able to improve both the performance and correctness of Memento creation and were able to significantly reduce the amount of transferred data and required processing time for Memento discovery through TimeMaps.

Our open source reference implementation of the proposed extension allows for the immediate evaluation of our proposal by the community and may serve as a foundation for future work. We conclude that in conjunction with our proposed extensions, the Memento protocol addresses a variety of data management challenges including data *archiving*, *citation*, *retrieval*, *discovery*, *synchronization* and *sustainability* for arbitrary and highly dynamic data on the Web and in Knowledge Graphs, providing a promising foundation for prospective standardized and interoperable data management solutions, e.g., in conjunction with the Linked Data Platform specification, which we plan to pursue in future work.

**Acknowledgments**  Funded by the Deutsche Forschungsgemeinschaft (DFG, German Research Foundation) under Germany's Excellence Strategy – EXC-2023 Internet of Production – 390621612.

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
