# OpenReview forum: "HTTP Extensions for the Management of Highly Dynamic Data Resources"
_eswc-conferences.org/ESWC/2021/Conference/Research_Track — ESWC 2021 Research_

### Official Review · AnonReviewer2 · 2021-01-11
**Clear, simple, and well-motivated Memento protocol extensions for highly volatile data**

**Rating:** 2
**Confidence:** 4
**Impact:** 3
**Design And Technical Quality:** 3

**Review:**

### Overview

This paper introduces three extensions to the Memento protocol to support persistence of versioned resources at high (sub-second) frequencies.
These extensions are described in detail, and the authors describe the reasons for designing them like this.
An open-source reference implementation is provided, on top of which the performance of ingestion and range-based lookups is measured.

The paper is clear and well written,
and the work is well-motivated, with a clear solution.
I especially applaud the authors on the discussion of backwards-compatibility,
as this is not always properly considered when designing such extensions.
The introduced extensions themselves are simple and straightforward,
which are a good thing the context of protocols.

Given the straightforward solution, I wonder if the research track is the proper target for this work.
Perhaps the resources track (or even in-use track given the clear use case) might have been better suited.
Nevertheless, I consider this work valuable, as it aims to solve a clear need.
Below, I list several points that could strengthen the paper.

### Point of improvement

The evaluation section is missing some crucial information.
First, it is unclear how large the created resources are, which may have a significant impact on performance.
Furthermore, varying resource sizes may also lead to varying performance differences.
Second, I am not sure if I understand the used network setup.
It is mentioned that a single server was used to run the Memento application.
It is however not clear where the client runs that executes Memento HTTP requests.
Does it run on the same machine, or on a separate machine?
If on the same machine, won't this impact the server load and possibly confuse the results?
If on a separate machine, what is the real/simulated network delay?

While the authors mention that their goal was not to provide a highly scalable implementation,
the paper is nevertheless missing a description of what their implementation looks like.
Even though the implementation is available on GitLab, an overview of its key architectural decisions should be mentioned.
This is important to understand the performance results,
which may possibly have more to do with the implementation than the protocol extensions.
For instance, the indexing strategy for range time maps is a crucial element to understand lookup times in figure 6.
Also, since Memento puts no restriction on resource size,
an in-memory implementation might not always be feasible.
Would these extensions be impacted in any way if the implementation
would persist to disk instead of fully working in-memory?

Even more general, the viability of a protocol extension depends a lot on the complexity of implementation,
as exemplified by the authors' mention of WebDAV.
However, a discussion on the implementation requirements for each protocol extension is missing.
For instance, what kind of datastructure would be good to use as back-end for a range request?
And what kind of algorithms would be needed to handle arbitrary resolution timestamps?

The related work section mentions works from the RDF versioning/archiving domain,
and concludes by saying that none of the existing approaches consider highly dynamic data sources.
Given the importance of highly volatile data, I would expect a mention of related work within the RDF Stream Processing community.
Especially, Triplewave [1] and Linked Data Notifications for RDF Streams [2] come to mind.

### References

[1] Mauri, Andrea, et al. "Triplewave: Spreading RDF streams on the web." International Semantic Web Conference. Springer, Cham, 2016.

[2] Calbimonte, Jean-Paul. "Linked Data Notifications for RDF Streams." WSP/WOMoCoE@ ISWC. 2017.

### After Rebuttal

I thank the authors for their rebuttal. Given the author's response to some of my questions, and theor promise to include RSP and additional implementation details in their final version, I have increased my overall score.

**Anonymity:**

No, I would like my review to be deanonymized.

**Reuse And Availability:**

4: High

**Strong Points:**

1. Clear solution to a real problem
2. Open-source implementation with reproducible experiments
3. Consideration for backwards-compatibility

**Subreviewer:**

I submitted this review.

**Weak Points:**

1. Evaluation section misses some necessary background information
2. Limited novelty
3. Missing impact discussion on implementation(s)

---

> ### Author Rebuttal · Authors · 2021-01-29
>
> Dear AnonReviewer2,
>
> thank you very much for your overall positive review and helpful remarks, which allow us to improve the quality of our paper further.
> In the following, we try to respond to each of the raised questions and points of improvement:
>
> * **Evaluation Setup:**
> As described in the introduction of section 4, the experiments are executed on a single workstation. While this certainly has an impact on the absolute performance achieved, as outlined in the paper, we did not focus on optimizing this metric. Instead, we compare the relative performance of two protocols under the same environmental conditions to highlight the relative performance benefit of the proposed protocol extensions, not the performance of our implementation.
>
> * **Implementation considerations:**
> We describe the basic architecture of our quite simplistic implementation in the beginning of section 4 and provide the code as open-source software. Detailed considerations, such as the choice of data structures, software parallelization considerations and employed persistence mechanisms, certainly have notable impact on the concrete performance profile of the implementation. This is also evident from the stark performance drop-off of our implementation at a more or less fixed throughput threshold observable in figure 5b, hinting at clear potential for optimization. However, they do not invalidate i) the general performance characteristics of the two protocol variants and ii) the functional advantages of the proposed extensions. Nevertheless, we commit to providing the requested additional information as part of the supplementary material of our final submission.
>
> * **Size of created resources in Evaluation:**
> The contents of the created resources are single random strings of length 20. Larger resources would effectively add the same transmission time offset for either protocol version, since this affects the initial PUT/POST transmission time equally for both protocol versions. Therefore, it is true, that the relative overhead of the second request diminishes as larger resources are transferred. Nevertheless, the absolute overhead will remain effectively unchanged, since an additional round trip is needed. We, however, do not deem this consideration relevant enough to motivate additional experiments, since the principal benefit gained – the avoidance of race conditions – remains completely unaffected. Nevertheless, we will include this reasoning in the final version of the paper.
>
> * **Impact of network latency:**
> Since we execute requests against localhost, the impact of network latency on our results is negligible. Increased network latency would impact the traditional Memento protocol (with one or multiple requests) equally or more than our proposal (with only a single request for all operations). We will also include this detail in the final paper.
> Finally, we would like to thank you very much for your time and effort to provide feedback on our submission, which allowed us to further improve our work.
>
> * **Novelty & Impact:**
> Similar to AnonReviewer3, and as argued in more detail in our response to AnnonReviewer4, we see clear novelty and a significant impact of our work for the unified management, persistent identification and archiving of highly dynamic data resources in the Web and as part of Semantic Data Management solutions. However, based on your feedback, we will further clarify these aspects as part of the final revision of our paper.
>
> * **Related work:**
> As we also discussed in our response to AnonReviewer5 in some more detail, RDF stream processing is indeed a relevant related topic, although focused on a different aspect of working with highly dynamic data resources, and we will incorporate a corresponding paragraph in the final version of our paper.
>
> Finally, we would like to thank you very much for your time and effort to provide feedback on our submission, which allowed us to further improve our work.
>
> Best Regards
>
> Lars Gleim, Liam Tirpitz, Stefan Decker

---

### Official Review · AnonReviewer3 · 2021-01-14
**Interesting work on broader Web technology**

**Rating:** 2
**Confidence:** 3
**Impact:** 4
**Design And Technical Quality:** 3

**Review:**

The authors present a proposal to extend the Memento Protocol to enable management of highly dynamic resources.

The paper is very well motivated, as it correctly points out that the original Memento protocol has shortcomings that prevent its use for important use cases for dynamic resources. It also motivates well that an extension of Memento can adequately address many such concerns, and that a relatively straightforward extension of Memento, which in itself is a relatively simple and quite well accepted extension to HTTP, and therefore bring very substantial benefits. The paper then goes on to detail such an extension. As such, it is a significant work.

I think the paper delivers on its promise, and that it does so with great clarity. In detail, I have some comments, but none that would alter the conclusions of the work very significantly.

Note, however, that I do not know the contemporary literature to any great extent beyond what is referenced in the paper, and therefore, I cannot make any strong evaluation of originality of the contribution.

The paper contains some simple experiments to verify its assertions. There is also a strong intuition behind the approaches, and so, the evaluation should not be surprising.

A case could be made against the relevance of the paper, as it does not address semantics to any significant degree. The present work is clearly broader than the Semantic Web, it applies to the Web in general, and would be usable with any highly dynamic resource. This is clearly a strength when looking at the paper in the bigger picture, but the Editors should consider whether this might disqualify the paper from being published on this venue. Nevertheless, I recommend acceptance.

After rebuttal:

Thank you very much for your rebuttal. The reviews have not changed my mind on your paper, I still recommend it for acceptance.

It is a good idea that I still think could have significant impact. Even though it isn't very deep on the research problems, it is significant as a contribution to solve a problem. I am assured that the technical objections I made can and will be solved partly before publication and partly in a subsequent standardisation process, which is appropriate.

As a comment for the Chairs, I still note that the paper would have found a broader and possibly more appropriate audience on the broader Web conference venues as it is not specialised as a Semantic Web contribution, but I feel that the Chairs are in a better position to take the big picture view than an individual reviewer to assess whether the paper fits in with the overall focus of the track as a whole or should be directed towards a general Web conference.


**Anonymity:**

No, I would like my review to be deanonymized.

**Reuse And Availability:**

5: Very High

**Strong Points:**

The paper makes an good case for benefits of extending the Memento protocol, and details a complete proposal for how key challenges can be met, and demonstrates that the proposal has many of the desired properties.

**Subreviewer:**

I submitted this review.

**Weak Points:**

The proposal is broader than the topic of the conference, which might be reason to redirect to a different venue. However, I think that most likely, the Semantic Web community will not only be the first adopters of the proposal, but very possibly also the main beneficiaries, and so I think it is appropriate to accept it.

My main technical objection to the paper is the proposal to use a `.well-known` URI as proposed in Section 3.3 when discussing an external Memento Archive. I don't find the justification for using a `.well-known` URI clearly articulated over other methods of discovery. I find it difficult to see the utility, because the client would have to know the URI-R anyway, and so you wouldn't be in a regime where very little knowledge is required.

Also, RFC 5785 says that:

> However, in keeping with the Architecture of the World-Wide Web [W3C.REC-webarch-20041215], well-known URIs are not intended for general information retrieval or establishment of large URI namespaces on the Web.

The paper seems to violate this guideline by suggesting that each resource should have a corresponding `.well-known` URI. If a `.well-known`  is used, it should be used merely to discover how the archive can be used.

I'm sure a different mechanism can be established without significantly alter the experimental results, and so, I don't think this problem is reason for rejection (this is the typical kind of problem that is addressed should the proposal make it into a standardisation effort, the authors should not be deterred).

The authors make a good case for why TimeMap ranges are required. However, I would also like to propose to the authors to consider whether it is appropriate to introduce an entirely new HTTP header for this purpose, or if it could be satisfied by the extension mechanisms provided by RFC 7233. Specifically, could it be satisfied by a new range unit? Again, such a change would not alter the experimental results, and so is not reason for rejection.

If, for some reason, the authors need to elaborate on some parts of the paper based on other reviews, I would suggest looking into shortening the discussion around Fig 2, perhaps even omitting it, as the authors make an excellent case for why the `Memento-Datetime` header should not only be a response header, but also a request header on write requests. It might suffice to say that without the ` Memento-Datetime` header on write requests, there is insufficient information to dereference a certain Memento. I found the caption of Fig 2 slightly confusing, even though the intuition behind it is very clear.

---

> ### Author Rebuttal · Authors · 2021-01-29
>
> Dear AnonReviewer3,
>
> Thank you very much for your positive and constructive feedback!
>
> The intention of using a `.well-known` URI is to provide a storage endpoint for Mementos with an external URI-R without inhibiting the same server to host resources with local URI-Rs. Based on your well-founded objection to the use of the `.well-known` URI as the storage endpoint itself, this mechanism can instead be used for the discovery of a concrete storage endpoint under a different path. We will adapt our proposal accordingly for the final submission. Further entirely different discovery mechanisms could certainly be conceived as part of a potential standardization effort.
> We further agree, that the existing RFC 7233 `Range` header should be adopted as part of a potential future standardization effort. However, we refrain from adopting it as part of this work, since this would entail the definition of a new datetime range unit, as you already pointed out, and does not affect the validity of our results.
>
> Finally, we would like to thank you very much for your time and effort to provide feedback on our submission, which allowed us to further improve our work.
>
> Best Regards
>
> Lars Gleim, Liam Tirpitz, Stefan Decker

---

### Official Review · AnonReviewer4 · 2021-01-14
**Modest contributions and insignificant impacts on Semantic Web**

**Rating:** 1
**Confidence:** 5
**Impact:** 2
**Design And Technical Quality:** 3

**Review:**

The paper proposed HTTP extensions to HTTP Memento protocol with the motivation from industrial IoT setting  which I personally is not convinced  due to the fact that HTTP PUT and POST of Memento  have  significant overhead in terms of communication payload and round trips. I wonder whether  authors  have considered or compare HTTP/2.0 or QUIC for the sake of fast throughputs.

For the paper does not have any scientific contribution. And the technical contributions are quite modest as the modifications are not very challenging. It's read more like a technical report than a research paper. Authors argue that HTTP Memento protocol play important role for the Web and Knowledge Graph but I don't see any convincing arguments to clarify the paper's direct contributions to semantic web research and technologies.  In fact, authors mentioned some previous work on RDF data management that used Momento protocol, but it not clear to me how the paper's technical contributions are aligned with the conference call around technical and research aspects of semantic web.  I think the paper is more suitable to the general Web conference like The Web Conference, The Web Engineering or Hypertext.


**Anonymity:**

Yes, I would like my review to remain anonymous.

**Reuse And Availability:**

4: High

**Strong Points:**

The paper published the implementation

**Subreviewer:**

I submitted this review.

**Weak Points:**

- The motivation is debatable

- The scientific and technical contributions are weak

- Relevancy and impact of the work on Semantic web research are not convincing.

---

> ### Author Rebuttal · Authors · 2021-01-29
>
> Dear AnnonReviewer4,
>
> Thank you very much for your feedback.
>
> * **Motivation, Relevancy & Impact:**
> Semantic Data Management (SDM) is an important aspect of the Semantic Web and a unified mechanism for the management, revisioning and persistent identification of arbitrary Web and Semantic Web resources is missing to date.
> This topic has recently been raised, e.g., in the context of the vision session of ISWC 2020 and received positive feedback from the community.
> We identify the problem that while SDM and versioning solutions in the Semantic Web community are largely using SPARQL, LDP or plain HTTP as their primary access mechanism, there is no standard for managing time series data and dynamic resources that would allow to handle RDF data analogously to any other resource on the Web.
> Especially in the context of IoT (even with current approaches such as the Web of Things family of standards) a joint and standardized mechanism for (time series) data management is missing.
> The Memento protocol has already repeatedly been successfully employed for time-based resource access in the context of SDM and with our proposal, we believe that it could provide the basis for interoperable solutions tightly integrated with the HTTP protocol and therefore the core technologies of the Web.
> While our paper may also be suitable for submission to a general Web conference, we believe that it is this (in our eyes) important missing connection between SDM, Web technologies and (I)IoT requirements, that motivates our submission to ESWC.
> Similar to AnonReviewer3, we further believe that the community may be the first adopter and have the most to gain from the proposal, since (as you suggested) there are arguably more “efficient” ways to exchange data then via HTTP (or RDF for that matter).
> Therefore, we submitted this paper to the “Semantic Data Management, Querying, and Distributed Data” track of the ESWC with the specific aspects “Distributed infrastructures for Semantic Data” and “Semantic data management techniques for [...] (FAIR) Data” from the CFP in mind. We strongly believe our work to be relevant w.r.t. those topics.
>
> * **Fast Throughput of HTTP/2.0 & QUIC:**
> In general, we agree that HTTP requests are associated with a certain overhead compared to other, potentially more efficient protocols that may be used for access to highly dynamic data.
> However, the Semantic Web (as well as the Internet at large) relies heavily on the HTTP protocol. Since our primary goal was an interoperable protocol that can be used across the Web in combination with arbitrary resources, we consider HTTP a good fit. Similarly, the Memento protocol is already in use, especially in Semantic Web applications and Web archiving, since it already provides a standardized way to revision arbitrary Web resources.
> Therefore, we build upon that work to maximize interoperability across the Web and provide a unified revisioning and persistent identification mechanism for arbitrary resources on the Web, even in highly dynamic application scenarios such as IoT.
> Even when more efficient protocols are employed for data collection, our proposal could still be used to facilitate the retrieval and identification of data and its revisions through the HTTP Memento protocol a.
> The current Memento protocol is indeed already compatible with HTTP/2.0 and QUIC (on the transport layer) and therefore the proposed extensions would also be compatible.
> Nevertheless, the core contribution of our proposal is the optimization of an existing and practically used Web protocol, while the further optimization of possible implementations of this scheme remains future work.
>
> Finally, we would like to thank you very much for your time and effort to provide feedback on our submission.
>
> Best Regards
>
> Lars Gleim, Liam Tirpitz, Stefan Decker

---

> > ### Comment · AnonReviewer4 · 2021-02-02
> > **fair and reasonable arguments**
> >
> > I would like to thank authors for addressing my concerns, I think authors offer fairly reasonable arguments so I will raised the score to weak accept.

---

### Official Review · AnonReviewer5 · 2021-01-14
**Review - HTTP Extensions for the Management of Highly Dynamic Data Resources**

**Rating:** 1
**Confidence:** 4
**Impact:** 3
**Design And Technical Quality:** 3

**Review:**

In this paper, the current Memento Protocol is extended to account for Highly dynamic data resources as found in the Industrial Internet of Things. The authors introduce new request and response headers as well as the usage of another datetime format to eliminate the identified shortcomings of the status quo.

The paper has a few shortcomings when it comes to the details of the solution (see points below). More relevant however, many state of the art technologies are not even mentioned, which might indicate that several important aspects of recent works are not reflected in the proposed contribution. Furthermore, the evaluation is not sufficient for the conference. This makes it impossible for the reader to estimate the added value of the extended Memento protocol.

## After Rebuttal

I want to thank the authors for their detailed and comprehensive comments. Even though I still have the feeling that the design of the evaluation determines the presented results rather than the design decisions of the protocol, the promised updates and clarifications to the text certainly justify its presentation at the conference. Therefore, I agree with the other reviewers and have adjusted my score.

**Anonymity:**

Yes, I would like my review to remain anonymous.

**Reuse And Availability:**

4: High

**Strong Points:**

- Time-related data is a significant gap in the IIoT, in particular the standardization consortia IIC, PI4.0 and so on do not provide sufficient recommendations yet. Using HTTP and Memento as an already defined protocol seems to be a valid idea, which needs to be discussed in the community. IIoT solutions tend to propose specialized protocols, which in the end make the system integration even harder.
- Good explanation of the shortcomings of the current protocol.
- The proposal of content negotiation through HTTP headers is a vey interesting approach for filtering the representations. Especially in combination with the outlined understanding of what a resource is – and how representations are slightly different – the overall suggestion looks like a very clean and transparent solution.
- Backward combability was implemented and illustrated nicely.


**Subreviewer:**

I submitted this review.

**Weak Points:**

- The related work section misses many relevant publications, or only presents the preliminaries to understand the previous Memento protocol. IoT and Linked Data/RDF Streams had a lot of attention in the recent years but are not mentioned at all. Mature standards like the LDP recommendation only appear in one sentence, even though it specifies many best practices that could help the operation semantics of the protocol. Outside the Semantic Web community, JSON Time Series are also a very wide-spread solution with many successful applications using it. Without mentioning these technologies, the reader cannot understand if and why the contribution is better than the state of the art.
- The operation semantics for the added HTTP verbs (POST/PUT) is not sufficiently discussed. For instance, the LDP recommendations makes a very interesting differentiation between them using LDP Containers. Using the Containers may also solve the problem with ‘original’ (on the origin server) and ‘remote’ ones (on a cloud/edge server). In particular, it prevents to use the original URI as a fragment of the used endpoint. This type of implicit information in the identifiers is a rather dangerous pattern, as systems could parse the identifiers and try to interpret them. Explicit statements using sameAs or seeAlso are certainly more transparent and less error-prone. Furthermore, I would ask the authors to analyze the idempotence characteristic of the introduced operations. Especially as a lot of representations are possible for a single resource, this promises relevant insights.
- The conducted experiments do not sufficiently support the original claims. The correctness of insertions for the classic Memento protocol must be <100% as soon as more than 1 request per second is sent, and must decrease exactly following the measured graph (Fig. 5a). This is not due to any advantage of the Memento 2 protocol but rather how Memento 1 is defined. The performance benefit of Fig. 5b is completely explained by the design of the experiment (Memento 1 client needs to make a second look-up, while this information is available through the response to the Memento 2 client). Similarly, the measured processing time of the second experiment is determined by its design (number of requests x duration of one request). The experiments therefore cannot explain why the design decisions are appropriate.
It must be said that it is in general very hard to evaluate a protocol. Maybe a more qualitative approach and comparison with the mentioned technologies above can provide more insights.
- some missing information: definition of time gate (considering reader has no previous knowledge of Memento) and differentiation of external time gates,
- no further explanation why the protocol can be “applied to the Linked Data versioning approach presented by Meinhardt et al. [9]”. Because its backwards compatible?
- some stylistic and grammatical issues: some very long sentences, some confusion with singular and plural verbs, some vocabulary issues i.e. “as is the case in our use case”

---

> ### Author Rebuttal · Authors · 2021-01-29
>
> Dear AnonReviewer5,
> thank you very much for your helpful and constructive feedback, which helped us to improve our revised paper and incorporate additional related work. In the following, we would like to individually address your concerns in detail:
>
> * **Related work regarding RDF Streams:**
> While our proposal focuses on the versioning, persistent identification and archiving of arbitrary highly dynamic data resources (which may thus be considered time series data), stream processing considers the analysis, usage and processing of a stream of data points, which may or may not include time series data.
> While we do certainly see applications where both technologies would be applicable, neither is directly dependent on the other.
> Nevertheless, we agree that the current attention put on stream processing technology further contributes to the subsequently greater relevance of our work for highly dynamic data resources. Therefore, we will include an extended discussion of this relationship and the related work in the final version of our paper.
>
> * **JSON Time Series (JTS) & other data formats for sharing time series data:**
> The Memento protocol provides a more general approach, that creates immutable and persistently identifiable revisions of arbitrary, dynamic Web resources and specifies a corresponding HTTP-based access mechanism for individual resource revisions/Mementos.
> The JTS data format could indeed be used instead of the Memento protocol when only concerned with the exchange of time series data. While this is a much more restricted application scenario, it is relevant related work and we will also discuss it in the final version of our paper.
>
> * **LDP:**
> Indeed, the combination of LDP and Memento is of great practical use. See our [recent paper](http://dbis.rwth-aachen.de/cms/publications/2021-btw-factstack/at_download/pdfFile).
> While the LDP indeed provides interesting primitives for the discovery and differentiation of directly controlled vs archived resources, we consider the semantic and implementational complexity of the LDP protocol to be too great to justify the inclusion of LDP containers into our proposal, since this effectively requires the implementation of large parts of the W3C LDP specification, significantly increasing complexity compared to the current IETF Memento protocol.
> Nevertheless, we consider this a promising direction for future work and would like to discuss the idea with you at ESWC 2021.
>
> * **Idempotence:**
> Our proposal maintains HTTP idempotence of all requests.
> While range request and increased timestamp accuracy clearly have no impact on idempotence, this question is interesting for resource creation:
> `Creating Mementos through the Original Resource` creates a new Memento with user-provided content and server-assigned `Memento-Datetime` for each PUT/POST request. It is therefore idempotent w.r.t the resource state itself, even though the header is updated.
> `Storing Mementos in a third-party Memento Archive` is always idempotent, both w.r.t. the resource state and its Memento header.
> Therefore, from the viewpoint of the “non-Memento-aware” Web, all described methods are idempotent.
>
> * **Evaluation Design:**
> While individual protocols are hard to evaluate, performance comparisons of different protocols for similar applications are quite common (e.g. TCP vs QUIC).
> Our empirical evaluation demonstrates that the advantages of the proposed extensions are not only intuitively plausible, but can be achieved and replicated in a real implementation. Therefore, our comparative, empirical evaluation fully supports the hypothesized theoretical performance benefit of our extensions.
>
> * **Definition of Time Gate:**
> While we do already describe & differentiate TimeGate and external TimeGate on page 3, we will improve the clarity of these concept definitions for the final paper, since they are indeed fundamental to the understanding of our proposal.
>
> * **Applicability to Meinhardt et al. [9]:**
> Exactly due to the backward compatible approach described in Section 3.3. To respond in detail: As summarized in our related work section, Meinhard et al. employ a custom REST endpoint for adding data to a Memento archive server. Section 3.7 of that paper outlines that this endpoint accepts state updates as full RDF graphs submitted via an HTTP PUT request. This endpoint further fulfills the described requirement that individual resource revisions are uniquely identified. Subsequently, this endpoint could be extended to implement the Memento creation scheme which we propose in section 3.3 of this work, directly returning Memento headers for PUT/POST requests.
>
> Finally, we would like to thank you very much for your time and effort to provide feedback on our submission and we hope that we were able to clarify the raised questions in a manner that allows you to reconsider an improved rating of our paper.
>
> Best Regards
>
> Lars Gleim, Liam Tirpitz, Stefan Decker

---

### Decision · Program_Chairs · 2021-02-23

**Decision:**

Accept

**Comment:**

After the rebuttal, the reviewers agreed that this work presents a valuable contribution to the Research Track of ESWC 2021. The reviewers welcomed the clear description of the proposed Memento extension, the consideration for backward compatibility, and the reproducibility of the results as the implementation is openly available. Therefore, this paper is recommended for acceptance.

For the camera-ready, we kindly ask the authors to carefully address the concerns and further comments raised by the reviewers. More precisely, we recommend the authors to:

1. Extend the related work to include the references pointed out by the reviewers.

2. Revise the application of well-known URIs in the approach.

3.  Strengthen the value of the proposed extension to the area of Semantic Data Management. This would address the concerns about the relevance of this work to the conference.